# Aerodynamic Performance Analysis of a Modified Joukowsky Airfoil: Parametric Control of Trailing Edge Thickness

**Pan Xiong** [1,2]**, Lin Wu** [1,2,*]**, Xinyuan Chen** [1,2,*]**, Yingguang Wu** [3] **and Wenjun Yang** [1,2]

1   Key Laboratory of Metallurgical Equipment and Control Technology, Wuhan University of Science and Technology, Ministry of Education, Wuhan 430081, China; whxpmechanical@gmail.com (P.X.); Guowt-wust@outlook.com (W.Y.)
2   Hubei Key Laboratory of Mechanical Transmission and Manufacturing Engineering, Wuhan University of Science and Technology, Wuhan 430081, China
3   Troops 32382 PLA, Wuhan 430081, China; wygwust@gmail.com
*   Correspondence: wulin618@hotmail.com (L.W.); chenxinyuan@wust.edu.cn (X.C.)

**Abstract:** In order to ensure the blade strength of large-scale wind turbine, the blunt trailing edge airfoil structure is proposed, aiming at assessing the impact of the trailing edge shape on the flow characteristics and airfoil performance. In this paper, a Joukowsky airfoil is modified by adding the tail thickness parameter $K$ to achieve the purpose of accurately modifying the thickness of the blunt tail edge of the airfoil. Using Ansys Fluent as a tool, a large eddy simulation (LES) model was used to analyze the vortex structure of the airfoil trailing edge. The attack angles were used as variables to analyze the aerodynamic performance of airfoils with different $K$-values. It was found that when $\alpha = 0°$, $\alpha = 4°$, and $\alpha = 8°$, the lift coefficient and lift–drag ratio increased with increasing $K$-value. With the increase in the angle of attack from $8°$ to $12°$, the lift–drag ratio of the airfoil with the blunt tail increased from +70% to $-7.3\%$ compared with the original airfoil, which shows that the airfoil with the blunt trailing edge has a better aerodynamic performance at a small angle of attack. The aerodynamic characteristics of the airfoil are affected by the periodic shedding of the wake vortex and also have periodic characteristics. By analyzing the vortex structure at the trailing edge, it was found that the value of K can affect the size of the vortex and the position of vortex generation/shedding. When $\alpha = 0°$, $\alpha = 4°$, and $\alpha = 8°$, the blunt trailing edge could improve the aerodynamic performance of the airfoil; when $\alpha = 12°$, the position of vortex generation changed, which reduced the aerodynamic performance of the airfoil. Therefore, when designing the trailing edge of an airfoil, the thickness of the trailing edge can be designed according to the specific working conditions. It can provide valuable information for the design and optimization of blunt trailing edge airfoil.

**Keywords:** Joukowsky airfoil; blunt trailing edge; LES; modification; aerodynamic performance

## 1. Introduction

With the continuous improvement of wind turbine power, wind turbine blades are becoming longer and longer. In order to ensure the strength of the blade, the trailing edge of the blade often uses a blunt trailing edge airfoil of large thickness [1–3]. Larger flow separation occurs at the trailing edge of the blade during operation, which reduces the efficiency of wind energy capture by the wind turbine. Therefore, knowing how to accurately predict and effectively improve the aerodynamic performance of the wind turbine airfoil has very important practical significance. Processing the trailing edge of the airfoil is one of the effective ways to improve aerodynamic performance. At the same time, many scholars have conducted much research on the thickness of the airfoil trailing edge.

The main methods of thickening the airfoil trailing edge are the direct truncation method [4], the symmetric thickening method [5], the asymmetric thickening method [6–8], and the airfoil rigid rotation method [9]. Hoerner [10] conducted a wind tunnel test study

on a blunt trailing edge airfoil obtained by direct truncation of a Gottinggen-490 airfoil. It was found that the maximum lift coefficient of the airfoil could be increased by increasing the thickness of the trailing edge, but this method will change the relative thickness of the airfoil or change the centerline distribution of the asymmetric airfoil. In this case, the modified airfoil and the original airfoil are not comparable. It is difficult to clearly show that the thickness of the blunt trailing edge of an airfoil changes its aerodynamic performance. Deng et al. [11] cut a DU97-W-300 airfoil directly after 79.3% of the chord length and extended the chord length to 1.0 in proportion to the truncated airfoil. It was found that the aerodynamic performance of the modified airfoil was worse than that of the original airfoil, indicating that not all airfoils modified to blunt trailing edge airfoils by direct truncation show improved aerodynamic performance. Standish et al. [12] symmetrically increased the thickness of an airfoil on both sides of the airfoil mid-curve through an exponential mixing function method that does not change the maximum thickness of the original airfoil and the centerline distribution. Chow et al. [13] found that blunt trailing-edge modification can not only provide a number of structural advantages, such as increasing the cross-section area and inertia moment of bend for a given maximum thickness and chord, but it also produces a great improvement in the lift coefficient and reduces the sensitivity to surface soiling. Baker [14] analyzed airfoils with a symmetric blunt trailing-edge thickness through the experimental method. The research results indicated that a moderate trailing edge thickness increase could increase the lift–drag ratio and reduce the leading-edge roughness sensitivity. Almohammadi et al. [15] studied the effects of four different blunt trailing edge shapes on the performance of wind turbines and found that circular and blunt trailing edge airfoils have an important impact on the performance of wind turbines. Jae-Ho Jeong et al. [16] found that using a blunt trailing edge airfoil at a specific position at the root of a large wind turbine can improve the inner force of the blade to a certain extent; compared with a pointed trailing edge wind turbine, the axial power of the whole blade is increased by 1%, and the bending moment of the blade is reduced by 0.5%. Jamieson [17] proposed that increasing the absolute thickness of an airfoil during blade design can increase the flapping stiffness of the section. Many scholars have conducted a lot of other research on airfoil trailing edges [18–20]. However, most studies were modified for a single airfoil, and there was no parameter control for modification of the thickness of the trailing edge, which cannot achieve the purpose of precise modification.

In this paper, a Joukowsky airfoil [21,22] is used as the original airfoil, which can obtain the desired target airfoil using an analytical formula. We added the trailing edge parameter *K* to control the thickness of the trailing edge of the Joukowsky airfoil to achieve the purpose of precise modification and used Ansys Fluent to analyze the influence of different *K*-values on the aerodynamic performance of the airfoil. Furthermore, the large eddy simulation (LES) approach was adopted to resolve time-wise and space-wise variation in the flow due to large-scale eddies.

## 2. Joukowsky Airfoil Function Modification

The lines of a blade or airfoil are usually determined by a coordinate database to give a series of coordinate point data; then, the image dot matrix is connected with a smooth curve in order to generate an airfoil profile line. In this way, a set of coordinate data only represents one airfoil [23,24]. The analytic function (Joukowsky airfoil transformation) is used to express the airfoil or blade. When the constant value in the function changes, a new airfoil can be generated, and the geometric meaning of the function parameters is very clear. By adjusting the parameter values, the desired airfoil can be generated, and a reverse design can be realized.

Draw two circles tangent to point B on a plane. The origin of the coordinate is the center of the big circle O, and the center of the small circle $O_1$ is on the line OB (Figure 1a). Complex variable analytic function:

$$\xi = \frac{1}{2}\left[z - \frac{c_0^2}{z}\right]; \ (z = \varsigma + i\eta; \ c_0 = r_0). \tag{1}$$

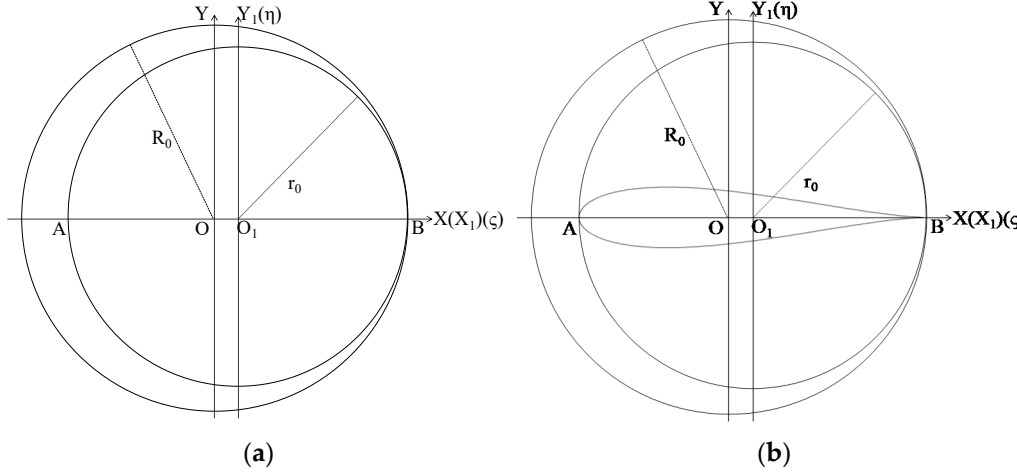

**Figure 1.** Joukowsky airfoil. (**a**) Step I. (**b**) Step II.

Conformal transformation of large circumference lines:

$$\varsigma = \frac{X_1}{2}\left(1 + \frac{c_0^2}{X_1^2 + Y_1^2}\right); \ \eta = \frac{Y_1}{2}\left(1 - \frac{c_0^2}{X_1^2 + Y_1^2}\right) \tag{2}$$

$$X_1 = X - (R_0 - r_0); \ Y_1 = Y. \tag{3}$$

Substitute the coordinates $(X, Y)$ of each point on the circumference of the large circle into the formula to calculate the coordinates of each point of the new airfoil $(\xi, \eta)$ (Figure 1b). The value of $r_0$ determines the chord length; $R_0 - r_0$ determines the thickness of the blade.

We thus obtain a symmetrical airfoil with a sharp tail. In order to change the thickness of the trailing edge, we need to improve the Joukowsky airfoil and move the large circle O along the $OO_1$ direction. As shown in Figure 2a, where $l_{BC}$ is the moving distance of the big circle, other parameters remain unchanged, and conformal transformation is performed on the large circle again.

$$X_1 = X - (1 - K) \times (R_0 - r_0); \ K = \frac{l_{BC}}{R_0 - r_0}; 0 \leq K \leq 1 \tag{4}$$

$$Y_1 = Y. \tag{5}$$

Substitute the coordinates $(X, Y)$ of each point on the circumference of the large circle into the formula again to calculate the coordinates of each point of the new airfoil $(\xi, \eta)$ (Figure 2b). Adjust the thickness of the trailing edge by adjusting different $K$-values. The $K$-values selected in this article are 0, 0.2, 0.4, and 0.6.

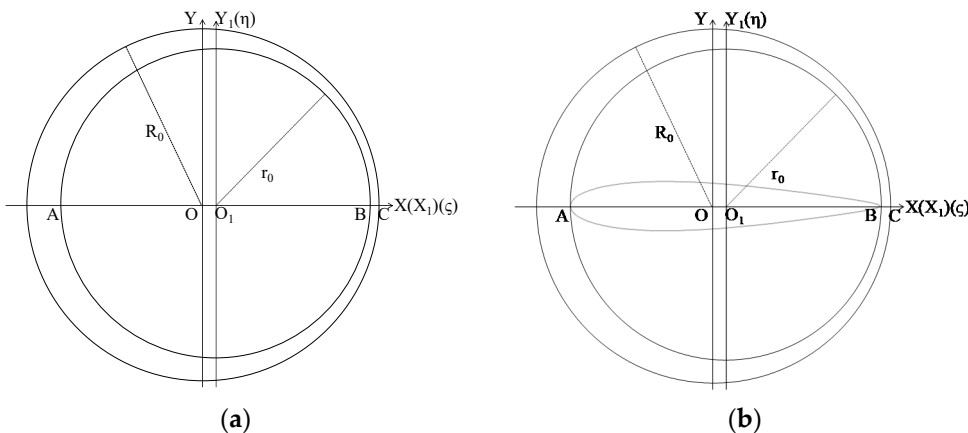

**Figure 2.** Airfoil tail optimization. (**a**) Step I. (**b**) Step II.

### 3. Calculation Model and Reliability Verification

This paper uses the size of Joukowsky airfoil studied by Modi [25] as the original size for research. The LES Smagorinsky SGS (sub-grid-scale) model approach was adopted to resolve time-wise and space-wise variation in the flow due to large-scale eddies. The finite volume method was applied to discretize the differential equation, and the SIMPLE algorithm was used to solve the discrete equation. The accuracy of spatial discretization was second-order. The Green–Gauss node-based method was used to calculate the gradient and derivative of variables at the center of the element cell through Ansys Fluent numerical simulation, and heat transfer was not considered. As two-dimensional calculations can also obtain results that are relatively close to experimental ones [26,27], the airfoil was calculated using a two-dimensional model. The model uses the Integrated Computer Engineering and Manufacturing code for Computational Fluid Dynamics (ICEM-CFD) for grid division, and the airfoil outer flow field radius is 20 chord. The fluid domain and fluid mesh is shown in Figure 3a. In order to ensure that the number of elements in the mesh has no impact on the results, a mesh independence verification was carried out, as shown in Figure 3b; when the number of meshes reached a certain value, further increases in the number of meshes had no significant effect on the calculation results but increased the calculation time. Considering the balance between solution accuracy and calculation time, the number of meshes selected for this study was approximately 0.4 million.

In order to ensure the accuracy of the simulation calculation, the Y+ value of each airfoil was less than 1 [28], as shown in Figure 4b. Additionally, in the LES, the Taylor length scale ($\lambda$) was examined to verify the sufficiency of the mesh resolution. According to Kuczaj [29], the mesh resolution ($\Delta = \sqrt[3]{V_{cell}}$) should at least be in the order of $\lambda$ to completely solve the Taylor length scale. This method was applied to the wake of an airfoil, at $Y/\delta = \pm 1$, since this is expected to be the area where the effects of the turbulence are most noticeable and, therefore, where the best resolution is required [30]; for data extraction, 11 spanwise planes normal to the streamwise direction located on the wake behind the airfoil were considered. These planes were located from $22\delta$ from the TE (trailing edge), separated $2\delta$ between each other, as shown in Figure 4a. Figure 4c shows that the criteria proposed by Kuczaj are fulfilled along the whole wake behind the airfoil, which means that the mesh is suitable for LES simulations. The details of the Computational Fluid Dynamics (CFD) model are presented in Table 1.

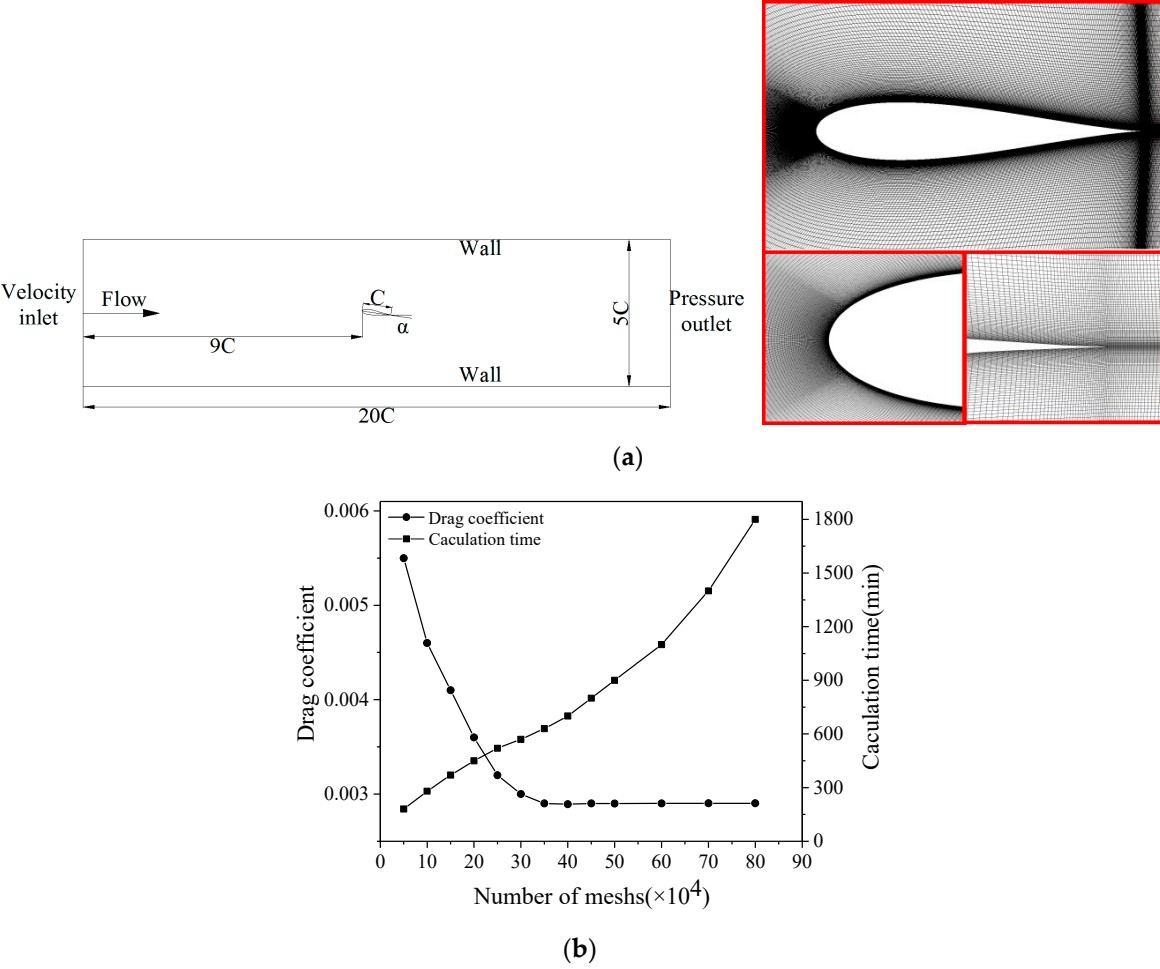

**Figure 3.** (**a**) Fluid domain and fluid mesh. (**b**) Mesh independence.

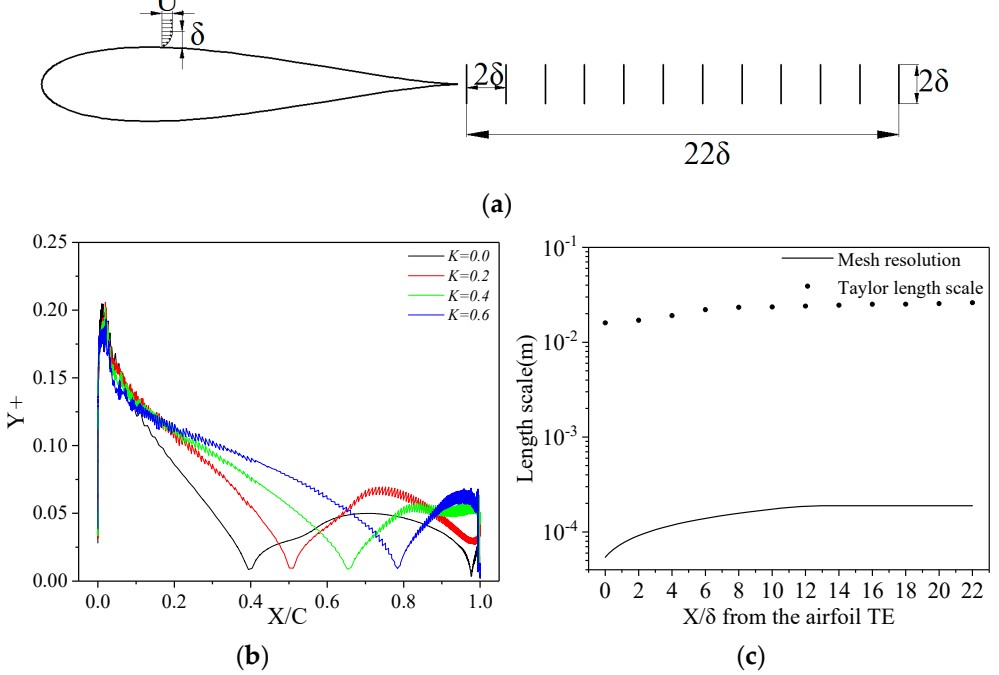

**Figure 4.** (**a**) Data extraction. (**b**) Y+ distribution of airfoil and (**c**) Taylor length scale on the wake behind the airfoil.

**Table 1.** CFD model parameters.

| Type | State |
| --- | --- |
| Boundary detail | No slip and smooth wall |
| Turbulence model | LES |
| Chord (m) | 0.38 |
| 1st layer thickness (m) | 0.00004 |
| Growth ratio | 1.1 |
| Maximum thickness (% chord) | 15 |
| Convergence absolute criteria | $10^{-5}$ |
| Inlet velocity (m·s$^{-1}$) | 1.618 |
| Angle of attack ($\alpha$) | $0°, 4°, 8°, 12°$ |
| K | 0, 0.2, 0.4, 0.6 |
| Number of grids (millions) | 0.38–0.40 |

In order to verify the reliability of this simulation, experimental data obtained from references were used [31,32]. The experimental data included the pressure coefficient distribution of the original Joukowsky airfoil at different Reynolds numbers and different angles of attack. The expression of the pressure coefficient is as follows:

$$C_P = \frac{p - p_{inlet}}{(0.5\rho U^2)_{inlet}} \tag{6}$$

where $p$ is the point static pressure on the airfoil; $p_{inlet}$ is the inlet static pressure; $\rho U^2$ is the dynamic pressure.

Comparing the simulation data with the experimental data, we found that the simulation data at different Reynolds numbers and different elevation angles are relatively close to the experimental data, which shows that the simulation has high reliability in analyzing the influence of different elevation angles and different Reynolds numbers on an airfoil, as shown in Figure 5.

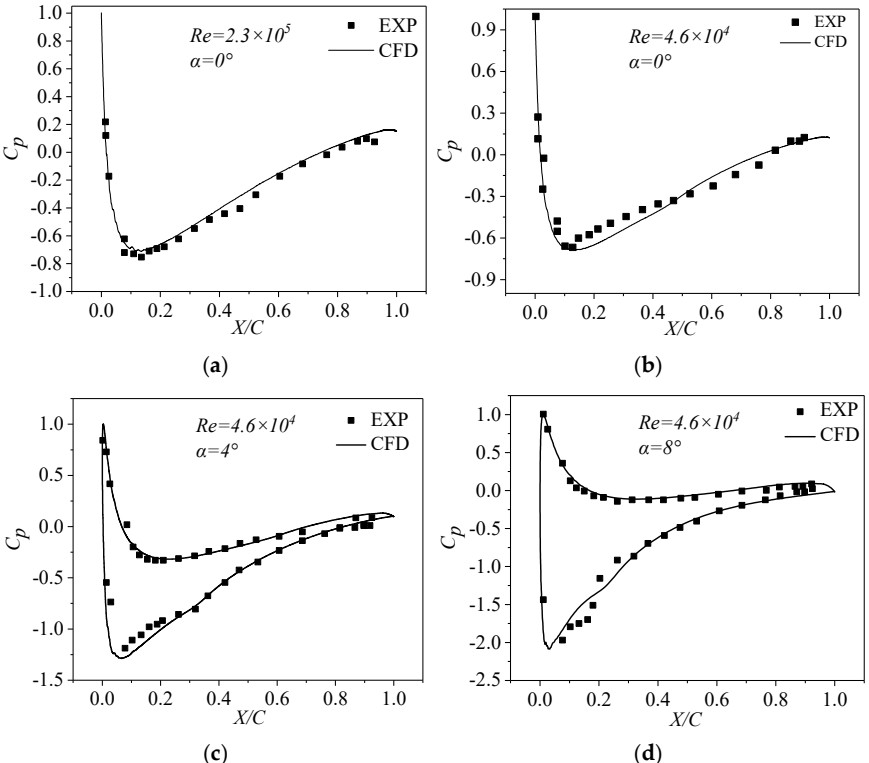

**Figure 5.** Comparison of simulation and experiment data [31,32]. (**a**) $Re = 2.3 \times 10^5$; $\alpha = 0°$. (**b**) $Re = 4.6 \times 10^4$; $\alpha = 0°$. (**c**) $Re = 4.6 \times 10^4$; $\alpha = 4°$. (**d**) $Re = 4.6 \times 10^4$; $\alpha = 8°$.

## 4. Aerodynamic Performance Analysis

Due to the unsteady flow field and the periodic shedding of the wake, the aerodynamic characteristics of the airfoil also show periodic characteristics. In order to ensure the accuracy of each working condition, when the lift coefficient and drag coefficient fluctuate periodically with time, the calculation can be considered to be convergent. In this study, the lift coefficient and the drag coefficient at 0.1 s after convergence of each working condition are intercepted for comparative analysis.

Figure 6 shows the aerodynamic performance distribution of the airfoil. Figure 6a,b show that, due to the symmetry of the airfoil, when $\alpha = 0°$, there is no pressure difference between the two sides, so the lift coefficient and drag coefficient will not change; when $\alpha \neq 0°$, the amplitude of aerodynamic performance increases with the increase of angle of attack, and the change of $K$ value has little effect on the frequency of aerodynamic performance. By comparing the average lift coefficient, the average drag coefficient, and the lift–drag ratio (Figure 6c–e, respectively), it was found that the drag coefficient decreases with the increase in K-value, which indicates that the drag coefficient of the blunt trailing edge airfoil can be well reduced. When $\alpha = 0°$, $\alpha = 4°$, and $\alpha = 8°$, the lift coefficient and lift–drag ratio increase with the increase in K-value; with the increase in the angle of attack from 8° to 12°, compared with the original airfoil, the lift–drag ratio of the blunt tail changes from +70% to −7.3%, showing that when the angle of attack reaches a certain value, the influence of the blunt trailing edge on the drag coefficient is greater than that on the lift coefficient, the slope of the drag coefficient increases, and finally, the slope of the lift–drag ratio decreases. With the increase in the angle of attack, the slope of the lift–drag ratio will decrease more. The greater the $K$-value is, the more obvious this phenomenon is. The vortex shedding frequency of the airfoil was determined by analyzing the Strouhal number ($S_t = f \times C/U$), as shown in Figure 6f. It can be seen from the figure that the $S_t$ number of the blunt tail is greater than that of the original airfoil at a small angle of attack. When $\alpha = 12°$, the $S_t$ number of the original airfoil is the largest, which is very similar to the distribution of the lift–drag ratio, indicating that the vortex shedding frequency is greatly related to the aerodynamic performance of the airfoil.

In order to further analyze why the influence of $K$-value on airfoil aerodynamic performance is completely the opposite when $\alpha = 8°$ and $\alpha = 12°$, we assessed vorticity for $K = 0$, $\alpha = 8°$; $K = 0.6$, $\alpha = 8°$; $K = 0$, $\alpha = 12°$; and $K = 0.6$, $\alpha = 12°$ at five selected moments (0 T, 0.25 T, 0.5 T, 0.75 T, and 1 T) in a stable cycle in each of the four working conditions, as shown in Figure 7. It can be seen from the figure that when the vortex on the suction surface (clockwise) is falling off, vortices on the pressure surface (anti-clockwise) are gradually generated and the lift coefficient increases gradually. As the vortex on the suction surface completely falls off, the lift reaches the maximum. At this time, the vortex on the pressure surface begins to fall off, vortices on the suction surface begin to form, the lift coefficient decreases gradually, and as the vortex on the pressure surface completely falls off, the lift coefficient reaches the minimum.

The aerodynamic performance of an airfoil is affected by the size of the vortex and the position of vortex generation/shedding. In order to analyze the influence of the airfoil with a blunt trailing edge on vorticity, the vorticity diagram of the airfoil trailing edge was analyzed, as shown in Figure 8. It can be seen from the figure that the value of $K$ affects the size of vortices and the position of vortex generation/shedding. When $\alpha = 8°$, the airfoil with a blunt trailing edge ($K = 0.6$) generates smaller vortices, and the vortices on the suction surface and pressure surface are located at the rear of the trailing edge (Figure 8b-⑤). The vortices generated by the trailing edge of the original airfoil ($K = 0$) are larger and are all near the suction surface (Figure 8a-⑤). The shedding/generation of vortices will have a greater impact on the aerodynamic performance of the airfoil. When $\alpha = 12°$, this phenomenon changed. The vortices generated by the $K = 0$ and $K = 0.6$ airfoils are near the suction surface, and the vortices generated by the $K = 0.6$ airfoil pressure surface are closer to the suction surface (Figure 8c-⑤,d-⑤). When the position of the vortices generated by the suction surface is the same (Figure 8c-①,d-②), the lift coefficient

is the same (Figure 7c-①,d-②). It was also found that the closer the vortices generated by the pressure surface are to the suction surface, the smaller the lift coefficient is. The lift coefficient increases with the clockwise vortex fall-off. As shown in Figures 7 and 8, when $\alpha = 8°$, the shedding position of the wake vortex with $K = 0.6$ is farther from the airfoil. With the increase in the angle of attack to $\alpha = 12°$, the shedding position of the wake vortex with $K = 0$ is farther from the airfoil. Under the same $K$-value, the shedding distance of the wake vortex decreases with the increase in the angle of attack. Under the same Reynolds number, the farther the shedding position is, the greater the shedding frequency is, which corresponds to the $S_t$ number in Figure 6f.

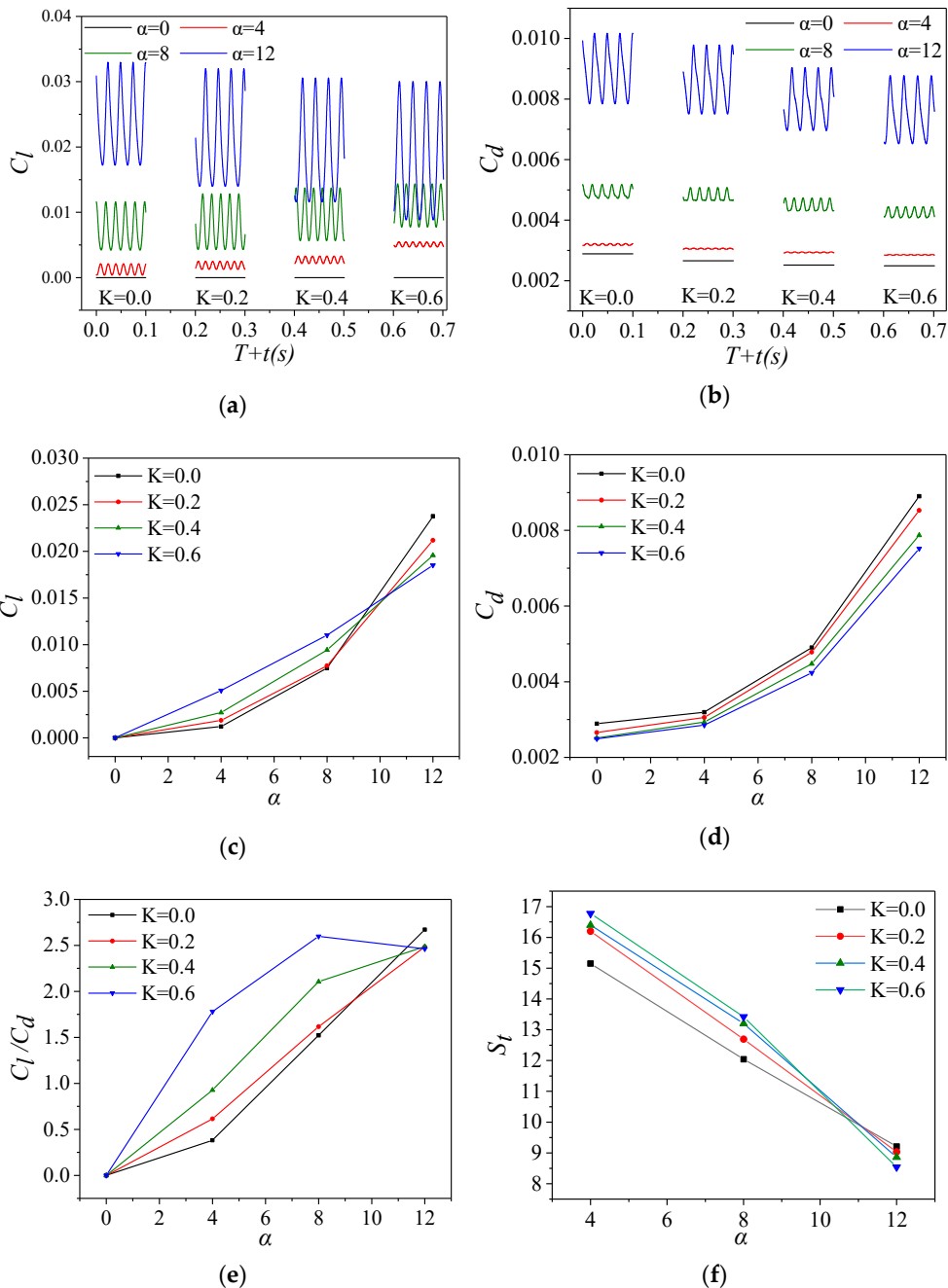

**Figure 6.** Aerodynamic performance of airfoil. (**a**) Time domain diagram of lift coefficient. (**b**) Time domain diagram of drag coefficient. (**c**) Lift coefficient at different angles of attack. (**d**) Drag coefficient at different angles of attack. (**e**) Lift–drag ratio at different angles of attack. (**f**) Strouhal number at different angles of attack.

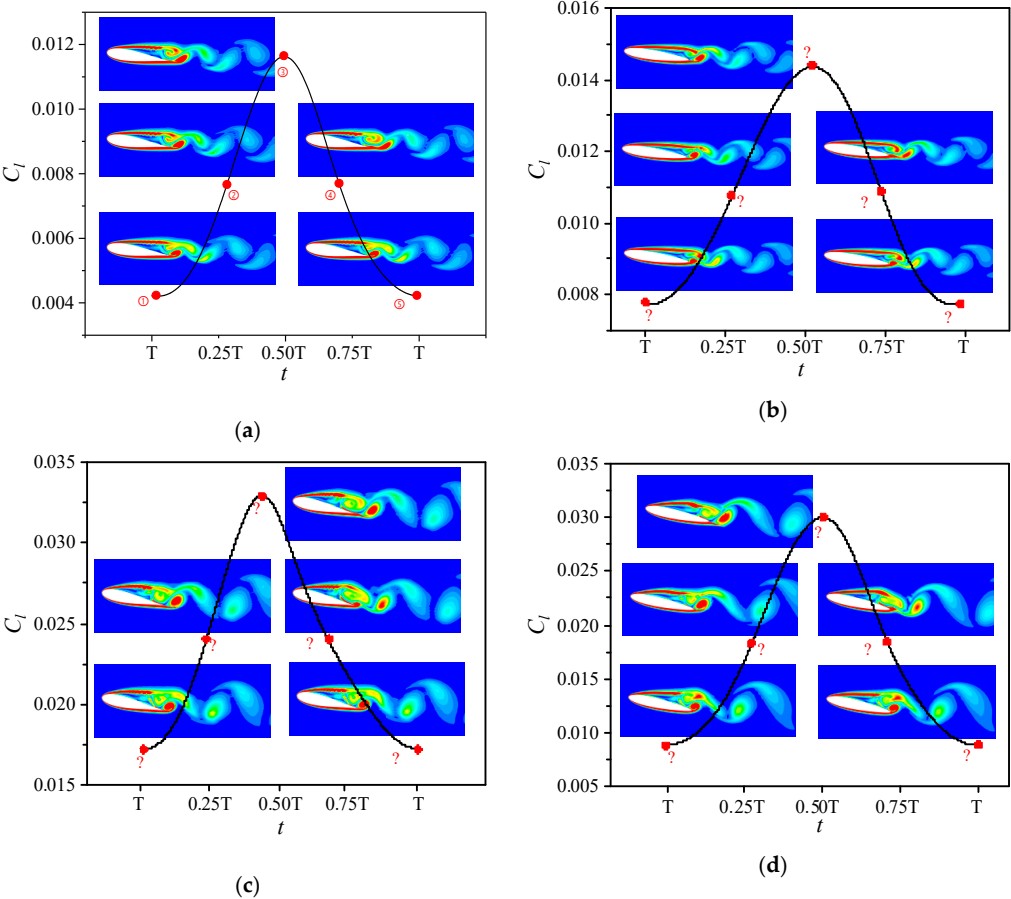

**Figure 7.** Vorticity diagram of a period. (**a**) $K = 0$, $\alpha = 8°$. (**b**) $K = 0.6$, $\alpha = 8°$. (**c**) $K = 0$, $\alpha = 12°$. (**d**) $K = 0.6$, $\alpha = 12°$.

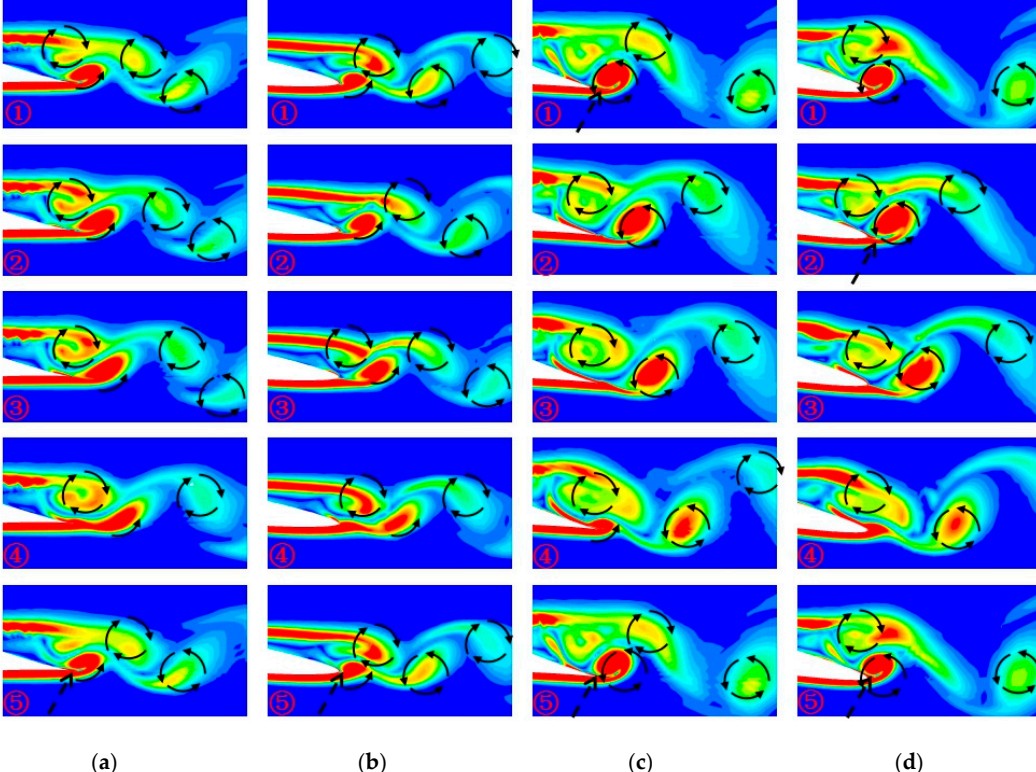

**Figure 8.** Vortex structure at trailing edge. (**a**) $K = 0$, $\alpha = 8°$. (**b**) $K = 0.6$, $\alpha = 8°$. (**c**) $K = 0$, $\alpha = 12°$. (**d**) $K = 0.6$, $\alpha = 12°$.

## 5. Conclusions

(1) By changing the Joukowsky airfoil and increasing the tail edge thickness control parameter K, the airfoil tail thickness can also be parameterized, making the blunt trailing edge more flexible and accurate in the Joukowsky airfoil. It is very convenient to optimize the aerodynamic performance of the airfoil by adjusting various parameters.

(2) The experimental results are in good agreement with the simulation results, indicating that the CFD model and simulation method are highly reliable.

(3) The aerodynamic characteristics of the airfoil are unsteady and periodic and are mainly affected by the size of vortices and the position of vortex generation/shedding. At the same angle of attack, the K-value can affect the size of vortices and the position of vortex generation/shedding. When $\alpha = 0°$, $\alpha = 4°$, and $\alpha = 8°$, the blunt trailing edge can improve the aerodynamic performance of the airfoil; when $\alpha = 12°$, the position of vortex generation changes, which reduces the aerodynamic performance of the airfoil. Therefore, when designing the trailing edge of an airfoil, the thickness of the trailing edge can be designed according to the specific working conditions.

(4) This article only proposes a parameterization method for the K-value to enable the optimization of a Joukowsky airfoil. No further specific analysis was performed. In the future, the optimal K-value can be obtained for specific working conditions to optimize the aerodynamic performance of airfoils.

**Author Contributions:** Conceptualization, P.X. and L.W.; methodology, P.X. and X.C.; software, P.X.; validation, Y.W. and W.Y.; resources, X.C.; data curation, P.X. and L.W.; writing—original draft preparation, P.X.; writing—review and editing, L.W.; supervision, X.C. All authors have read and agreed to the published version of the manuscript.

**Funding:** This research received no external funding.

**Institutional Review Board Statement:** Not applicable.

**Informed Consent Statement:** Not applicable.

**Data Availability Statement:** Data is contained within the article.

**Conflicts of Interest:** The authors declare no conflict of interest.

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
