# Peer review of "Aerodynamic Performance Analysis of a Modified Joukowsky Airfoil: Parametric Control of Trailing Edge Thickness"

_applsci, doi:10.3390/app11188395_

Round 1

Reviewer 1 Report

Journal Applied Sciences (ISSN 2076-3417)

Manuscript ID applsci-1320634

Type Article

Title Aerodynamic Performance Analysis of Modified Joukowsky Airfoil: Parametric Control of Trailing Edge Thickness (K)

A study of the thickness parameter k of Joukowsky airfoil is presented in the current work as the original airfoil. Computational LES based methods are used for the parametric study. The authors used a commercial code ANSYS Fluent.

The paper face an interesting subject and seems to be of interest for the scientific community, however some issues have to be addressed. An exhaustive English grammar and style revision is mandatory.

Abstract is well organized, but I miss more quantitative results.

The introduction is too short and does not cover all the state of the art on the paper subject. Only 15 references are not enough for a highly investigated topic such as TE optimization. 

Line 97 a ref is missing

Figure 3. The structured mesh seems to be nice and also the validation versus experiments with the cp is nice. But it is not enough to show that your results are reliable. I recommend the authors to taylor length. See my comments below.

Which type of LES model is used? SGS usually is recommended. In LES, Taylor length-scale (λ) has to be examined to verify sufficient mesh resolution. According to Kuczaj et al. [], the mesh resolution (Δ= √3[Vcell]) should at least be in order of λ to completely solve the Taylor length-scale. Taylor length-scale calculation procedure consists of obtaining the autocorrelation function from the Taylor expansion coefficient, then, calculating the Taylor time-scale, and finally, estimating λ from the Taylor hypothesis See the study of Portal-Porras et al DOI: 10.3390/pr9030503 for more information

Line 105: why Yplus has t be below 0.2?? it is totally new for me.

Figure 7 is nice. Very well done. In addition, Figure 8 is also nice but the discussion of those results has to be extended, elaborate it more.

Conclusions are well supported by the results.

Reviewer 2 Report

Dear authors,

I have read you article, and I am unfortunately not happy with it. Here are the reasons:

The context and motivations are not clear.

  • in the abstract, you do not mention your field of research ,
  • in the abstract and introduction, you keep talking about "thickening the trailing edge", but you do not explain why it is needed,
  • in section 2, you say that you will use the Joukowsky airfoil, but again, you do not explain why (you simply state that it can be parametrized),
  • etc.

The article is lacking some important content.

  • in section, the choice of the numerical aerodynamic model is not justified
  • in section 3, there is no domain sensitivity analysis,
  • in section 3, there is no grid sensitivity analysis,
  • in section 3, there are no details on the numerical model (LES), nor on the numerical schemes that have been used
  • in section 3, there are no details on the domain setup (boundary conditions, etc.)

I know you have compared your numerical results to experiments, but that does not mean that the results are accurate and that you reached domain/grid independence.

The English language must be improved. Most sentences are written in a weird fashion, and punctuation is often not correct. Some sentences are even meaningless.

I am not convinced by the overall scientific added value of the present work. You use a very detailed aerodynamic model (LES) to analyze an "academic" airfoil, impractical to wind turbine design. You also consider the effect of trailing edge thickness alone, while aerodynamic performance is a function of more than that. In a real application, optimizing an airfoil with a sharp TE, and then thickening it does not make sense. You would start with a thick TE airfoil instead, and optimize it as such.

Even though, the review highlighted many negative points, I still would like to mention that I liked the your scientific approach in section 4, where you tried to explain what was causing the increase/decrease in CL and CD.

I hope that you can improve your work, and publish it later, when it has reached a more mature state.

Sincerely,

Reviewer 3 Report

It is a nice piece of work that will be of good interest to aerodynamics community.

Authors investigated the effect of tip thickness on modified Joukowsky airfoil based on CFD analysis in detail.

I recommend this paper to be accepted after the following minor concerns are addressed.

1) There are various typographical/grammatical errors, e.g.

              Line 176: … Joukowsky airfoil airfoil and … (duplicating “airfoil”)

              Line 104: I think “Y+” with Y in capital letter is usually expressed in small letter as “y+”.

              etc.

Please proofread the manuscript.

2) I could not find any information about grid convergence. In such a CFD-based study, authors should provide convergence information, by using at least 3 different grids with different number of grid points.

3) Please also provide CFD scheme information: what was the spatial discretization method and its accuracy (2nd order or 3rd order in space)?

4) In Fig 5(b), Cp value shows discrepancy from experimental data by 10% at around x/C=0.2 and 0.7. Do authors have any idea about this reason?

5) What was the Strouhal number St found in your CFD simulation? Was it the same as St in past studies?

Round 2

Reviewer 1 Report

my comments have been successfully addressed

Author Response

English has been modified by professional institutions.

Reviewer 2 Report

Dear authors,

I am glad to see that your revised version of the paper has vastly improved compared to your first submission. I still have a few recommendations:

  • In your abstract, please mention the context (wind turbine blade design), and the motivation (for structural reasons, the TE needs to be thick, which decreases the airfoil's efficiency, hence the need of this study). This is basically what you explain in the introduction, but in a shorter form.
  • Please add a sketch of your domain including the boundaries and the boundary conditions, along side table 1. Clearly state what B.C. you used. Do not simply mention "inlet", you need to specify which variable or gradient is fixed, and to which value. Basically, any reader should be able to reproduce your computations by simply reading your text.
  • I would make it very clear that you do not try to optimize/design the full airfoil, but that you rather focus on the impact of the TE's shape on the local flow. I then makes sense to use such an airfoil (Joukowsky) with such a very detailed aerodynamic model (LES).

Sincerely,
